Occurrence, ecological function and medical importance of dermestid beetle hastisetae

Ruzzier Enrico enrico.ruzzier@phd.unipd.it symphyla@gmail.com 1
Kadej Marcin 2
Battisti Andrea 1
1 Department of Agronomy, Food, Natural Resources, Animals and the Environment (DAFNAE), Università degli Studi di Padova , Padova , Italy
2 Department of Invertebrate Biology, Evolution and Conservation, University of Wrocław , Wrocław , Poland
Gillespie Joseph
Electronic publication date: 2020 Jan 23
Publication date: 2020
Volume: 8
Electronic Location ID: e8340
Received 2019 Jul 1; Accepted 2019 Dec 3
Copyright: ©2020 Ruzzier et al.
Copyright year: 2020
Copyright holder: Ruzzier et al.
License: This is an open access article distributed under the terms of the Creative Commons Attribution License, which permits unrestricted use, distribution, reproduction and adaptation in any medium and for any purpose provided that it is properly attributed. For attribution, the original author(s), title, publication source (PeerJ) and either DOI or URL of the article must be cited.
License URL: https://creativecommons.org/licenses/by/4.0/

Keywords: Allergy, Coleoptera, Systematic, Zoology, Dermestidae, Health, Ecology, Insects

Funding: University of Padova, DAFNAE department, DOR FINA grants This work was supported by the University of Padova, DAFNAE department, DOR and FINA grants. The funders had no role in study design, data collection and analysis, decision to publish, or preparation of the manuscript.

==============================
Hastisetae are a specific group of detachable setae characterizing the larvae of Megatominae (Coleoptera: Dermestidae), commonly known as carpet and khapra beetles. These setae are located on both thoracic and abdominal tergites and they are the primary defense of the larva against invertebrate predators. According to previous studies, the main purpose of hastisetae is to work as a mechanical obstacle, but they are also capable to block and kill a predator. Hastisetae, single or aggregate, function as an extremely efficient mechanical trap, based on an entangling mechanism of cuticular structures (spines and hairs) and body appendages (antennae, legs and mouthparts). It is believed that this defensive system evolved primarily to contrast predation by invertebrates, however it has been observed that hastisetae may affect vertebrates as well. Although information on the impacts of vertebrate predators of the beetles is lacking, hastisetae have been shown to be a possible threat for human health as an important contaminant of stored products (food and fabric), work and living environment. Review of past and recent literature on dermestid larvae has revealed that despite these structures indicated as one of the distinctive characters in species identification, very little is known about their ultrastructure, evolution and mechanism of action. In the present work, we will provide the state of knowledge on hastisetae in Dermestidae and we will present and discuss future research perspectives intended to bridge the existing knowledge gaps.

Introduction

The cuticle plays a pivotal role in several aspects of arthropod biology, representing the interface between the living tissue and the external environment (Bereiter-Hahn, Matoltsy & Richards, 1984). Thus, the cuticle displays structural specializations such as denticles, setae, setulae and spines, all with specific functions (Winterton, 2009). Correlations between structure and function are well studied especially in insects (Neville, 1975) and crustaceans (Garm, 2004a; Garm, 2004b; Garm & Watling, 2013). Setae are multicellular protuberances on the cuticle, used primarily for mechanoreception (Keil & Steinbrecht, 1984; Keil, 1997; Winterton, 2009; Barth, 2004). In all groups of arthropods, the role of setae has evolved from simple mechanoreception to various other functions, including defense (Battisti et al., 2011), locomotion (Labarque et al., 2017), prey capture (Felgenhauer, Watling & Thistle, 1989), pheromone dispersal (Steinbrech, 1984), sexual display (Perez-Miles et al., 2005), preening (Felgenhauer, Watling & Thistle, 1989), and camouflage (Zeledón, Valerio & Valerio, 1973; Hultgren & Stachowicz, 2008). Detachable setae are true setae characterized by the loss of the neural connection and the detachment of the base of the hair from the integument (Battisti et al., 2011). The proximal end of each seta is attached to an integument stalk or inserted into a socket and can be easily removed with any kind of mechanical stimulation. This class of hairs has evolved as a defensive structure against predation at least four times in Arthropoda. The class is subdivided in two main morpho-ecological groups: urticating hairs and anchor-like setae. Urticating hairs are characterized in some Lepidoptera families such as the Nodotontidae (subfamily Thaumetopoeinae), Erebidae, Saturniidae and Zygenidae and the spider family Theraphosidae (subfamily Theraphosinae) (Battisti et al., 2011) and are described to protect from vertebrate predators (Battisti et al., 2011; Bertani & Guadanucci, 2013). Anchor-like setae are characterized in some larvae of Dermestidae (Insecta: Coleoptera) and Polyxenidae (Myriapoda: Polyxenida) where they work as entangling mechanism against invertebrates (Nutting & Spangler, 1969; Eisner, Eisner & Deyrup, 1996). Dermestid detachable setae (hastisetae) are used by the larvae as an active trapping system against arthropod predators (Nutting & Spangler, 1969). These specialized setae are almost exclusively prerogative of Megatominae, the most species rich group in the entire family (Háva, 2015). The mechanism of action of hastisetae and their microstructure remains largely obscure and restricted to few case studies (Nutting & Spangler, 1969; Mills & Partida, 1976); furthermore, how the evolution of hastisetae is related to the biological success of the Megatominae remains unresolved. Although information on the impacts of hastisetae on vertebrate predators is lacking, dermestid larvae and Megatominae in particular have been documented as possible source of allergens in human (Mullen & Durden, 2009). Hastisetae and integument fragments carrying them can be contaminants of stored commodities and are present in working and living environments (Hinton, 1945). Hastisetae seem to be involved in allergic reactions through skin contact, ingestion or inhalation; symptoms can vary accordingly to exposition and consist of skin rushes, asthma, conjunctivitis and digestive system inflammation (Gorgojo et al., 2015; MacArthur et al., 2016). Correlation between the presence of hastisetae and the incidence of allergies in humans exists but the scarce and incomplete information available do not allow to consider hastisetae as a major hazard in living and working places. The aim of this review is to synthesize the knowledge on the hastisetae of dermestid beetles, to evaluate their occurrence in the group and their ecological importance, and to assess their possible implications in the human health. Finally, future perspectives on the study of the hastisetae with special emphasis on Megatominae are envisaged.

Survey methodology

In order to compile and then review the most exhaustive literature on hastisetae we performed a careful and reiterated research in Google Scholar and Scopus through the use of keywords such as “hastisetae”, “Dermestidae”, “defense”, “larva”, integrated by the usage the Boolean operators AND, OR, NOT and the use of ” ” for specific word combinations. The literature not available online has been recovered thanks to Network Inter-Library Document Exchange (NILDE), a web-based software for the service of Document Supply and Inter-Library Loan, managed by the Italian National Research Council. Our research has enabled the collection of more than a hundred publications, of which ninety were considered in the realization of this review. The library created was comprehensive of literature in English, German and French.

Figure 1 Hastisetae structure and distribution on Megatominae larvae (general scheme).

(A). Example of Megatominae larva (Megatoma undata (Linnaeus, 1758)), dorsal view. T1–T3: thoracic segments; A1–A8: abdominal segments. (B). Tuft of hastisetae on abdominal segments. (C). Hastisetae, lateral view. (D). Head of the hastiseta (subconical anchor-like, spear-shaped head). Image credit: Paolo Paolucci, Michał Kukla.

Results

Hastisetae, structure and function

Hastisetae (or hastate setae) have been cited in several papers dealing with Dermestidae systematics (Rees, 1943; Kiselyova & McHugh, 2006), species identification (Booth, Cox & Madge, 1990; Peacock, 1993), and product contamination (Bousquet, 1990). However, the amount of information available concerning their microstructure (Elbert, 1976; Elbert, 1978), function (Nutting & Spangler, 1969; Mills & Partida, 1976) and evolution (Zhantiev, 2000; Kiselyova & McHugh, 2006) is quite scarce. These hairs, located on the dorso-lateral surface of the tergites of larvae and pupae (Fig. 1) (Rees, 1943; Beal, 1960; Kiselyova & McHugh, 2006; Kadej, 2012a; Kadej, 2012b; Kadej, Jaroszewicz & Tarnawski, 2013; Kadej & Jaroszewicz, 2013; Kadej & Guziak, 2017; Kadej, 2017; Kadej, 2018a; Kadej, 2018b), are generally quite small with an estimated length, according to the literature, between 150 and 900 µm. Density and distribution of the hastisetae vary substantially not only among genera and species but also among tergites of the same species. The hastisetae of the thoracic segments are generally scattered and in low numbers in respect to the other parts of the body. While the abdominal tergites present a wider distribution pattern, from hastisetae covering the major part the tergal disc up to proper setae fields located at the posterior corners of tergites (i.e., Reesa, Trogoderma). In some larvae, the hastisetae give origin to real tufts of hairs located on the posterior corners of the terga IV–VII (i.e., Ctesias) or V–VII (i.e., Anthrenus) (Mroczkowski, 1975; Kadej & Jaroszewicz, 2013; Kadej, 2017; Kadej, 2018a; Kadej, 2018b). The hastisetae are inserted in setal sockets on the integument and are connected to the tormogen cell trough the pedicel (Elbert, 1978). The pedicel is the breaking point of the shaft which allows the detachment of the hastiseta (Elbert, 1978). Hastisetae microstructure consists of two main parts: the shaft and the apical head (Fig. 1). The shaft is long and filiform, subcylindrical in section. It is made by repeated modules, from 5 to 77, each of them constituted by one cylindrical segment provided with one wreath of spines/scales in the distal part (Elbert, 1978). These spines/scales are postero-laterally oriented and can vary in number from five to seven (Elbert, 1978). The last module of the shaft is generally bigger and thicker than the previous and can slightly vary in general shape to the others; this structure, however, has not been characterized yet. The head of the seta is a subconical anchor-like, spear-shaped structure subdivided longitudinally in sections; the apex of the head is blunt (Elbert, 1976; Elbert, 1978) (Fig. 1). The head consists of five to seven longitudinal, circularly arranged, elements separated from each other by one deep groove, connected to the stem in the upper half by cross-bracing and free in the lower part. The “anchor-like head”, set against the thorns of the last shaft module, is involved in entangling invertebrate body parts (Nutting & Spangler, 1969), functioning as trap for antennae, legs, mouthparts, setae and spines (Mills & Partida, 1976). This structure is apparently species specific, varying in shape and length between taxa (Elbert, 1976; Kiselyova & McHugh, 2006; Kadej & Jaroszewicz, 2013; Kadej, 2017; Kadej, 2018a). The shaft allows setae to cluster together amplifying the “trapping” effect and the spines increases friction and entangling among hastisetae and between setae and body parts. The combined action of several hastisetae affects small predators (Nutting & Spangler, 1969) and possibly food competitors (Kokubu & Mills, 1980). These setae are hollow (Elbert, 1976; Elbert, 1978) and could potentially contain proteins or other chemicals involved in the defense, as it has been shown in Lepidoptera (Battisti et al., 2011). Hastisetae morphology and distribution, combined together with other characters, constitute a useful tool for species identification (Rees, 1943; Beal, 1960; Peacock, 1993; Kadej, 2012a; Kadej, 2012b; Kadej & Jaroszewicz, 2013; Kadej & Jaroszewicz, 2013; Kadej & Guziak, 2017; Kadej, 2017; Kadej, 2018a; Kadej, 2018b).

Figure 2 Schematic representation of Dermestidae phylogeny (based on Kiselyova & McHugh, 2006), with an indication of feeding habits of the adult beetles, duration of larval lifespan, and larval-pupal defensive structures.

The size of the colored bands in each subfamily is an approximated representation of the number of species. Image credit: Paolo Paolucci.

Hastisetae in the systematic and ecology of Dermestidae

Dermestidae is a cosmopolitan, comparatively small family of Coleoptera, regarded as ‘a well-defined, monophyletic group’ (Lawrence & Newton, 1982), consisting of six subfamilies: Orphilinae, Thorictinae, Dermestinae, Attageninae, Trinodinae and Megatominae (Háva, 2015) (Fig. 2). Dermestids are homogeneous only in general appearance, hiding a complex and rich diversity in term of morphological, ecological and ethological aspects. Specific traits and evolutionary tendencies could be observed in several lineages, associated to ecological groups and niches (Zhantiev, 2009); these traits can be observed at adult (Zhantiev, 2000) and larval stage (Kiselyova & McHugh, 2006). Orphilinae are mycetophagous, with sclerotized burrowing larvae (Lenoir et al., 2013). Thorictinae are myrmecophilous and larvae protection is provided by the associated ant species (Lenoir et al., 2013). Dermestinae, the basal group of the “necrophagous clade” (sensu Zhantiev, 2009), have larvae feeding on fresh or relatively humid animal remains (over 15% in water content) (Zhantiev, 2009). Since Dermestinae food resource is highly perishable, the larvae develop rapidly and persist only for short periods. The oblong, sub-cylindrical and sclerotized larvae of this subfamily can dig through the feeding substrate and live in butyric fermentation condition, under animal remains. It’s is plausible that the absence of hastisetae on larval tergites is directly attributable to their burrowing lifestyle. Anchor-like detachable setae could be disadvantageous to move within the substrate. Hastisetae would in fact create friction and would be systematically lost, requiring an important energy expenditure necessary for their replacement. The defensive strategy in Dermestinae is based on the fast escape behavior and the sclerotized integuments of the body. The larvae specifically require the pupation chamber to molt and they are capable to dig into soil and/or substrate in case of lacking suitable places where to hide. The pupae of this subfamily present gin-traps on the integuments, as a defensive system against predators (Hinton, 1946; Kiselyova & McHugh, 2006) (Fig. 2). Attageninae have burrowing larvae associated to wood dust, fissures of rocks and sandy environments and feed off of insects and other arthropods remains; the larvae are oblong-fusiform with integuments covered of three different kind of hairs (Zhantiev, 2000; Kiselyova & McHugh, 2006). The larvae show a fast escape behavior, similar to Dermestinae. Attageninae prefer to pupate in hidden niches and the pupae bear gin-traps in most of the cases (Zhantiev, 2000). Trinodinae are inquiline of animals’ nets: rodent borrows with larvae phoretic on mammal (Zhantiev, 2009) or larvae associated to spider nests (Beal, 1959; Kadej, 2012c). The hastisetae, with the single exception of the genus Trinodes (Trinodinae), in which modified hastisetae are described (Kiselyova & McHugh, 2006), are prerogative of the Megatominae larvae and they are strictly associated to larval and pupal morphology and behavior (Kiselyova & McHugh, 2006; Zhantiev, 2009) (Fig. 2). Megatominae is the richest in species subfamily within Dermestidae and its biological success is most probably attributable to the hastisetae occurrence. Amber fossils indicate that hastisetae morphology is highly conserved and remained virtually unchanged since late Cretaceous (Poinar Jr & Poinar, 2016). This group shows a remarked investment on hastisetae as a defensive tool (Nutting & Spangler, 1969; Mills & Partida, 1976), exploiting their resistance and durability over time to protect both larvae and pupae (Kiselyova & McHugh, 2006; Zhantiev, 2009). Megatominae is the clade within the xerophilous necrophagous dermestids (sensu Zhantiev, 2009), which can survive on low-water food resources, especially chitinous and keratinous remains (Armes, 1990; Beal, 1998; Zhantiev, 2009). These substrates are capable to stand in the environment for a long time but the poor nutrients prolong the duration of larval development, with major implications on morphology, ethology and defensive behavior. Lengthening of the larval phase and its persistence in the environment for a long time has promoted the evolution of morphological and ethological features in Megatominae that otherwise would have been disadvantageous in a different lifestyle. The inability of the larvae to delve into the living substrate (Zhantiev, 2009) favored the evolution of defensive structures (hastisetae) with low energy investment for their synthesis and to remain functional even after being dispersed in the environment. Over time, energetic investment in cuticularized integuments in larvae and gin-traps in pupae shifted to the morphology of hastisetae and its defense mechanisms. Hastisetae provide protection in both larvae and pupae, favoring a positive energy trade-off in larval development. All the larvae of this subfamily are stout, feebly sclerotized, slow moving and present an aggressive, non-escaping defensive behavior (Kiselyova & McHugh, 2006). In a disturbance, the larva stops moving, arches its body and spread the hastisetae, frequently from the posterior part of the body where it is densely packed with hastisetae towards the stimulus (Kiselyova & McHugh, 2006). In general, Megatominae do not make pupation chambers or hide, but simply pupate where they have been feeding. Pupae completely lack gin-traps and remain protected inside the last larval exuvia, completely covered in hastisetae (synapomorphy of Trinodinae + Megatominae) (Kiselyova & McHugh, 2006) (Fig. 2). Megatominae have been able to adapt against interspecific and intraspecific competition for food resources. A common trait associated with the evolution of the hastisetae in the dermestids is, in the necrophagous clade, the transition from scavenger habits of adults to anthophily or aphagy (Zhantiev, 2009) (Fig. 2).

Hastisetae and human health

The capability to feed on a wide range of food resources scarce in water content and to resist to prolonged starvation makes Megatominae larvae the perfect candidate to inhabit working and living spaces. In addiction, due to their slow movements and cryptic behavior these larvae result difficult to detect and remove. For this reason, some species are now synanthropic and cosmopolitan (Bouchet, Lavaud & Deschamps, 1996; Gamarra, Outerelo & Hernández, 2009), having been spread all over the world with trade. These species became serious pests, causing considerable loss and damage to stored goods of both animal and plant origin (Hinton, 1945; Burges, 1959; Kantack & Staples, 1969; Mroczkowski, 1975; Beal, 1991; Veer, Prasad & Rao, 1991a; Veer, Prasad & Rao, 1991b; Veer & Rao, 1995; Veer, Negi & Rao, 1996; Imura, 2003; Rajendran & Hajira Parveen, 2005; Lawrence & Slipinski, 2010) and to objects of organic origin in museums of cultural and natural history (Jurecka, Gebhart & Mainitz, 1987; Zaitseva, 1987; Armes, 1988; Bousquet, 1990; Pinniger & Harmon, 1999; Stengaard et al., 2012; Querner, 2015). The hastisetae released by the larva throughout its entire development and abandoned in the environment in association to the exuviae are an important contaminant in dwelling, public spaces as well as food stuff (Gorham, 1979; Gorham, 1989; Burgess, 1993) and can contribute as allergens in humans (Wiseman et al., 1959; Johansson, Wüthrich & Zortea-Caflisch, 1985; Baldo & Panzani, 1988; Burgess, 1993; Pauli & Bessot, 2009; Gorgojo et al., 2015; MacArthur et al., 2016): chitin, likely the main constituent of the hastisetae, is in fact a powerful and widely recognized allergen, and its interaction with Th2 lymphocytes and human chitinases enhances the inflammation process (Brinchmann et al., 2011; Bucolo et al., 2011; Mack et al., 2015). However, it is still unclear whether the inflammatory effect of the hastisetae is attributable to the mechanical action of the seta and its penetration through the epithelia or if it is associated to the presence of specific molecules capable to start an immunological reaction. Hastisetae have been directly linked to occupational diseases in working environments (Loir & Legagneux, 1922; Renaudin, 2010), especially when processing organic materials such flour, wool, silk and other commodities (Veer, Negi & Rao, 1996; Brito et al., 2002), or stored objects of organic origin in museums and art galleries (Siegel et al., 1991). The exposure to and inhalation of hastisetae, even in the form of dust, are reported to cause rhinoconjunctivitis (Brito et al., 2002) and asthma (Cuesta-Herranz et al., 1997; Brito et al., 2002; Bernstein et al., 2009). Megatominae are also one of the arthropod groups most commonly recorded inside houses (Gamarra, Outerelo & Hernández, 2009; Bertone et al., 2016; Madden et al., 2016); the larvae persist in these environments for months, even for years, feeding on food (Gorham, 1979; Gorham, 1989; Hirao, 2000), pet food (Rudolph et al., 1981), dust, insect remains and clothes, especially wool fabric (Bouchet, Lavaud & Deschamps, 1996). This prolonged presence inside houses together with the persistence of the hastisetae in the environment greatly increase the possibility for the humans to come into contact and develop a sensitization to these detachable hairs (Wiseman et al., 1959; Ayres & Mihan, 1967; Kaufman, Bado & Tovey, 1986; Burgess, 1993; Jakubas-Zawalska et al., 2016). The direct exposure of hastisetae to the skin, maybe due to contaminated bed or clothes, causes severe dermatitis (Sheldon & Johnston, 1941; Cormia & Lewis, 1948; Okumura, 1967; Ahmed et al., 1981; Alexander, 1984; Johansson, Wüthrich & Zortea-Caflisch, 1985; Southcott, 1989; Horster et al., 2002; Zanca, Zanca & Cassisa, 2012; Hoverson et al., 2015; MacArthur et al., 2016), while the repeated inhalation over a longer period may cause asthma (Cuesta-Herranz et al., 1997; Brito et al., 2002; Bernstein et al., 2009). Food contamination and hastisetae ingestion has been proved to cause the inflammation of the digestive system, manifesting through nausea, fever, diarrhea (Hirao, 2000), proctitis and perianal itching (Krause et al., 1998). Unusual, and apparently asymptomatic findings of hastisetae have been done on sputum (Johnson & Batchelor, 1989) and cervical specimens (Bechtold, Staunton & Katz, 1985; Bryant & Maslan, 1994; Williamson, Nicolas & Nayar, 2005). The incidence of pathologies associated with Dermestidae and Megatominae in particular, seems to be considerably reduced in recent decades probably due to the increased degree of attention regarding the presence of contaminants in food and the marked improvement in the processes of conservation and storage of raw materials; the development of adequate plans for monitoring and management of pests and the general improvement in the quality of life of people associated with greater healthiness of the houses have contributed further to the imitation of the impact (Athanassiou & Arthur, 2018). However, there is also the possibility that many domestic cases of exposure to hastisetae, especially in the case of skin rushes (erythematobullous reactions) may be under-recognized and underdiagnosed, due to similar effects to attacks by other arthropods (Burgess, 1993; MacArthur et al., 2016). Furthermore, almost all the cases reported in the medical literature regard developed countries while the effect of hastisetae on human health in developing countries remains almost obscure and widely understudied. Undoubtedly, a better knowledge of the inflammation caused by hastisetae would allow the recommendation of appropriated prevention measures and the formation of medical personnel able to provide early diagnosis and administration of appropriate therapies. Moreover, a close collaboration between occupational physicians, entomologists and immunologists could be of great help for the development of new surveillance programs and new health and safety guidelines for workers and people most at risk.

Conclusions

The scant information about the fine morphology and the ecological roles of hastisetae, and their implications in human health opens a whole horizon of research possibilities. Hastisetae morphology is undoubtedly the starting point for any future study. The characterization of hastisetae through electron microscopy and micro-CT is the basic and fundamental step to understand their functional morphology. The identification of specific morphological traits in the hastisetae will help to solve Megatominae systematics, highlighting the evolution of these structures in relation to phylogeny and biology. A detailed knowledge of hastisetae morphology will allow us to understand the defensive mechanism and if it acts similarly in all Megatominae. Comparing reactions of different predators to hastisetae will be useful to evaluate the different effects and particularly if this defensive system is primarily directed towards invertebrates and/or to vertebrates. Are humans or other vertebrates possible targets of hastisetae, and if so what are the causes of the unpleasant side-effects in humans? Is it the penetration of these setae trough epithelia the main cause of inflammation and are there any particular substance inducing the reaction, as it has been showed in Lepidoptera? Chemical analysis of secretions can identify and characterize the compounds responsible of the inflammation in humans and clarify their possible role as adjuvants in defense against the threats. Understanding the causes of allergic responses in humans will allow the development of specific medical therapies. Hastisetae could become an important addition in species identification, with relevant application in forensic entomology and pest management on stored products. Furthermore, the creation of a molecular fingerprint based on hastisetae content can aid in developing tools to detect insect fragments in contaminated stored products, especially food.

Thanks to Mizuki Uemura (Università degli Studi di Padova) for language editing, to Paolo Paolucci (Università degli Studi di Padova) and Michał Kukla (University of Wrocław) for images realization, to Antonio Masi (Università degli Studi di Padova) and three anonymous referees for useful suggestion provided during manuscript realization.

Additional Information and Declarations

Competing Interests

Author Contributions

Data Availability

The authors declare there are no competing interests.

Enrico Ruzzier and Andrea Battisti analyzed the data, conceived and designed the experiments, performed the experiments, prepared figures and/or tables, authored or reviewed drafts of the paper, approved the final text.

Marcin Kadej analyzed the data, performed the experiments, prepared figures and/or tables, authored or reviewed drafts of the paper, approved the final text.

The following information was supplied regarding data availability:

This is a literature review which did not generate any data. All data used are cited in the references.

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
