# Peer review of "Occurrence, ecological function and medical importance of dermestid beetle hastisetae"

_PeerJ, doi:10.7717/peerj.8340_

## Round 0.1 · original submission · Major Revisions

Dear Dr. Ruzzier and colleagues:

Thanks for submitting your manuscript to PeerJ. I have now received three independent reviews of your work, and as you will see, the reviewers raised some concerns about the review. Despite this, these reviewers are optimistic about your work and the potential impact it will have on research pertaining to dermestid beetle biology. Thus, I encourage you to revise your manuscript, accordingly, taking into account all of the concerns raised by the reviewers.

While the concerns of the reviewers are relatively minor, this is a major revision to ensure that the original reviewers have a chance to evaluate your responses to their concerns.

I look forward to seeing your revision, and thanks again for submitting your work to PeerJ.

Good luck with your revision,

-joe

Reviewer 1 ·

Basic reporting

This paper is basically a review of the occurrence and function of hastisetae in dermestid beetles.

Experimental design

there is really no study design since no new information is added, just a survey of past information. Full citation of the articles discussed is provided.

Validity of the findings

the topic is interesting and if anything points out new areas of research that can be organized with dermestid beetles.

Additional comments

The authors should make it clear that hastisetae are also produced by members of the subfamily Anthreninae, as well as by members of the subfamily Megatominae.

In figure 1, the genus and species of the example of the Megatominae larva should be provided. It certainly reminds me of a species of Anthrenus, which belongs to that subfamily.

Can Figures 1B and 1C be made with a bit more contrast? Especially Figure 1B.

Reviewer 2 ·

Basic reporting

The review is written well with clear English and a clear goal for the reader. The context is important and has applications in both the stored product pest discipline and medical discipline (allergens). The literature surveyed was exhaustive and their search criteria seem reasonable. The review is broad enough to be of interest to several groups of people, and the authors offer some interesting follow-up experiments that would be of particular interest in bio-control. It seems like the review cites several semi-recent articles that describe different functions of the setae but there are no recent reviews. The introduction is lacking a bit of background information. I would suggest including more information on the reason studying these setae matter, as mentioned in the abstract. The authors mention that the hastiseate can be “a serious threat to public health” but don’t provide any more details than that. I’m not convinced that statement is as powerful as they want it to be without further evidence. Although the authors do go into more detail on this subject within the review, so possibly mentioning that it will be explored further is important.

Experimental design

I think the phylogeny portion and the description of the structure and function are done well and with sufficient detail. The survey methodology is sufficient to find appropriate literature on their subject and they present their findings in an organized manner. Source information is presented well and it seems like they found a variety of sources from evolutionary to pest control. Subsections are organized logically and easy to follow. I might follow-up each section with a short sentence or two on what possible experiments could be done to expand the knowledge of this section and why would expanding what we know be important. It seems like the authors just present the information and, although the provide some conclusions, could provide more context as to why this review matters as the manuscript moves along.

Validity of the findings

Conclusions are presented well and provides unresolved questions and gaps that can be answered by future research. I think the thing that is missing for me is the well-developed and supported argument that meets the Introduction goals. I don’t think the introduction sets up how important these species are and how important understanding the evolution and physiology of these hastisetae are for public health and stored product protection. There is nothing that says understanding these three factors (phylogeny, health, and structure) will provide us with information on how to protect people from certain things in the future. How will this help in pest control? How will this help in preventing allergic attacks? Figures are very nice and provide a nice look at the structure and phylogenetic context of the hastisetae. They are much needed and appreciated for understanding this review.

Additional comments

Overall, I don’t have any specific comments besides to make sure you make the “Why” of the review very clear and to emphasize at the end of each subsection of the results why these results are interesting and how they can be applied to future directions. I think you have provided a nice look at the current state of knowledge but the future and why this matters is so important.

Reviewer 3 ·

Basic reporting

The goal of this paper, which meets all criteria for a scientific review, is to summarize all data available on the ecological and medical aspects of certain cuticular structures (hastisetae) of the dermatid species Megatominae.
The article is well structured, follows a clear ductus and highlights the function of hastisetae for the defence of these slow moving insects which are feeding on nutritients with low water content.
The authors provide a comprehensive list of references.
However, the reader would expect one or two more figures (e.g., life cycle, adult stage).
Please, add scale to image of larva.
The language is highly professional. Only the meaning of the sentence starting with "Based om amber fossils,..." (see. Results 170 - 172) is not clear to the reviewer.

Experimental design

Classical review, authors make methodology of literature research clear to the reader.
The information given is very good as long as it meets enthomological aspects of the issue.
Unfortunately, the (possible) importance of Megatominae larvae for mankind and human health is not elaborated in the same precise manner.
Some phrases are too "catchy", e.g., "serious threat for public health", "it is now widely recognized that these hairs have important and unpleasant effects on humans".
Most of the medical references presented by the authors had been published before 2000, some report only on single cases.
The authors should find arguments why there is not more interest on this issue in the medical literature.
"Public health" means hazard to the community. Who are the persons at risk; is it / might it become a potential occupational disease, etc.
The imbalance between the excellent enthomological part of the paper and the medical section is obvious. The medical part is significantly less than a page whereas the enthomology stretches over almost four pages.

Validity of the findings

No comment

Additional comments

The reviewer would like to encourage the authors to revise the medical part and present more details on possible pathogenetic mechanisms. Do hastisetae penetrate mucous membranes or skin? They are so different from true setae of Lepidopt. which have an arrow-like appearance that it is difficult to imagine that hastisetae really penetrate the skin barrier. What are the hints for toxic ingredients located withinthe cuticular structure?

---

## Round 0.2 · Minor Revisions

Dear Dr. Ruzzier and colleagues:

Thanks for revising your manuscript. The reviewers are very satisfied with your revision (as am I). Great! However, there are a few minor edits to make. Please address these ASAP so we may move towards acceptance of your work.

Best,

-joe

Reviewer 1 ·

Basic reporting

.

Experimental design

.

Validity of the findings

.

Additional comments

The revision meets my approval.

Reviewer 2 ·

Basic reporting

Much improved manuscript. I think the authors have enhanced their review substantially adding significant amounts of literature and developing their arguments fully. Basic reporting as categorized is done well.

Experimental design

No comments.

Validity of the findings

The arguments are much better defined and supported by the literature. You have made the importance of studying and understanding the hastisetae much more clear. The conclusion nicely wraps up all of your previous arguments and makes a case for further research.

Additional comments

Just a few general comments and edits:
Line 30: Change old literature to past or something similar.
Line 33: Remove "of the art" so it is just "the state of knowledge"
Line 45: Structure and function are well studied
Lines 62-67: These sentences seem to be mostly just fragments. Consider combining them or just give a definition without the "While" or "whereas" phrases.
Line 82: Do not allow what?
Line 103: Edit for punctuation
Line 127: "Apex blunt" is an odd phrase. Are you missing a word?
Line 163: "defensive strategy Dermestinae" are you missing a word here?
Line 169: Feed off of
Line 172: Phrasing of "are usually present gin-traps" is odd
Line 198: Punctuation. "aggressive, non-escaping"
Lines 205-210 are redundant to information presented above. Consider moving above or deleting
Line 216: Odd phrasing. Consider revising

Reviewer 3 ·

Basic reporting

Due to the revision paper has now gained the required quality level for publication in PeerJ.
Entholmological section and medical section are now well balanced in length and scientific content.
However, there are some issues which should be clarified before definite publication.

Experimental design

No further comment

Validity of the findings

What does authors make think that there is a toxic content in the cavity of setae? Own experience? Literature?

The authors should be aware that in the section Hastisatae and human health they write about "allergic" phenomenons. Toxic means everybody will be affected, allergic means it is bad luck that someone responds to an allergen.

Section Hastisatae and human health

Do the authors really mean "...prolonged inhalation...." or "repeated inhalation/exposure over a longer period"?

"...can determine the insurgence of asthma..." is a very uncommon "medical" formulation. Better: "may cause asthma"

Authors should make clear that in every manifestation with suspected allergic background prevention is the most important step. Allergic symptoms as well as treatment are all the same independet of causal pathogenesis.

An image depicting life cycle of the insect is still missing.

Additional comments

Quality of paper has increased significantly. If you add the few issues mentioned above the paper will be an excellent contribution to PeerJ.

---

## Round 0.3 · accepted · Accept

Dear Dr. Ruzzier and colleagues:

Thanks for re-submitting your revised manuscript to PeerJ, and for addressing the concerns raised by the reviewers. I now believe that your manuscript is suitable for publication. Congratulations! I look forward to seeing this work in print, and I anticipate it being an important resource for research communities studying dermestid beetle biology.

Thanks again for choosing PeerJ to publish such important work.

-joe